# k-Odd One Clear (k-OOC), a novel GPU kernel that improves quantization accuracy and speed of GPTQ algorithm

## Abstract

Large Language Model (LLM) demonstrated tremendously useful applications in nowadays fast-evolving AI driven technology. As the model sizes grow bigger, the demand for bigger and faster GPU is required. Another way to alleviate this issue is by improving the compression of the trained model through quantization so that lower VRAM devices can run. Quantization paradigms like GPTQ, PB-LLM, BiLLM (Hessian based with structural searching) are successful quantize mechanisms. In this paper, we propose **OOC**, a technique to pick an "odd" group to improve the quantization clarity so that the model can have better reasoning capability overall. In addition, we define **Bit Family** ($A^{lim}, A^{max}$) to classify compression rate of current and past quantizing techniques, thus providing a more objective way to rank different methodologies in literature. Thirdly, to avoid compromising the quantization speed due to the **scanning** process overhead, we developed a specialized fused GPU kernel (k-OOC) where it can be $9\times$ faster than the original GPTQ implementation (single-flow mode) and $22\times$ faster than the naive OOC implementation (double-flow mode) due to the incorporation of techniques called **Row-Flow-Selection Parallel** and **Input Batching**. We measured perplexity (PPL) of k-OOC (2 bits) with 14 major models like OPT, LLAMA, and Bloom (125M to 70B parameters) and popular datasets ( Wikitext2, C4, and PTB). We managed to improved the PPL of small model by 8.9% and of big model by 4.1% compared to the baseline of GPTQ (2 bits).

## 1 Introduction

Popular successful LLM models are often based on transformer architecture (Vaswani, 2017). If only considering the Full Precision at Float 16 (4 bytes) per weight, some of those models like OPT (Zhang et al., 2022; Radford et al., 2019), LlaMA (Touvron et al., 2023), and BLOOM (Le Scao et al., 2023) can reach 60-70 billion parameters, costing more than 100GB just to load the models onto the GPUs. It is a legitimate need to compress these models using quantization (popularized by (Dettmers et al., 2022)). A natural approach is model compression like in the work of Hoefler et al. (2021), however, methods like Quantization-Aware Training (QAT) and Post-Training Quantization (PTQ) are more favorable because of its inference quality. PTQ is trending more because it is one-shot and does not require any grad calculation (back-propagation). A few notable PTQ works are done by Nagel et al. (2021); Nahshan et al. (2021); Yao et al. (2022). Some PTQ methods are based solely on the curvature of the Hessian Matrix (Frantar & Alistarh, 2022; Frantar et al., 2022; Huang et al., 2024; Yuan et al., 2024), and the idea of calculating the salient metric from Hessian Matrix of each column in a weight matrix $W$ dates back to the second-order model pruning techniques (Hassibi et al., 1993; LeCun et al., 1989) and recently improved upon by (Frantar et al., 2021; Yu et al., 2022). After quantizing a model, the new weights are normally bench-marked against some datasets using the perplexity (PPL) metric (Arora & Rangarajan, 2016). Wikitext2 (Wikipedia articles by Merity et al. (2016)), C4 (web-scraped English passages by Raffel et al. (2020)), and PTB (Wall Street Journal articles by Marcus et al. (1993)) are of the most relevant sources.

A compression rate is a critical metric that classifies and evaluates different techniques, and can be measured by how many bits on average are used to store the information of a quantized weight matrix. However, this concept has not been formalized or taken into account holistically in previous

works, and only briefly mentioned Huang et al. (2024); Yuan et al. (2024). This leads to the case where some works "incorrectly" claimed to achieve a lot lower compression rate than what the actual rate is. In this work, we introduced 3 main contributions: a) define **Quantization Bit Family** to comprehensively classify compression rate, based on the observation that those rate can be mathematically estimated ($A^{lim}$, $A^{max}$ in section 3.1), regardless of the model/ layer/ modules size; b) Based on that framework, GPTQ with 2 bits per weight is currently one of the lowest compression rate with $A^{lim} = 2.1$, thus we aim to create a technique called OOC to improve the PPL score while maintaining in the same $A^{lim}$ family; c) from the insight that each row of the weight matrix $W$ can be quantized independently, we introduce the kernel version of OOC, called k-OOC, that speed up the original GPTQ, and also help to deal with the additional cost of running OOC.

## 1.1 RELATED WORKS

In the realm of large model quantization, three major and recent PTQ works that are good for benchmarking against are GPTQ (Frantar et al., 2022), PB-LLM (Yuan et al., 2024), and BiLLM (Huang et al., 2024). GPTQ is an efficient quantization method that can quantize the large models like OPT-175B in ∼4 hours, and can quantized with 3 or 4 bits per weight without affecting the original PPL too much. With the goal of exploring how far a model can be compressed, this work bench-marked against GPTQ most extreme regime of its variation, GPTQ(2) for 2 bits per weight. There were a few ways that a list of number $A = [a_1, a_2, a_3, ..., a_n]$ can be quantized. The simplest method is Round-To-Nearest (RTN) (Yao et al., 2022; Dettmers et al., 2022) or the *sign* method where $f_{\text{sign}}^q(a) = \text{sign}(a) \times \text{scale}, \forall a \in A$, where a typical choice for *scale* is scale=$\overline{|A|}$. [1] Another way is to use *GPTQ quantization using $n$ bits*, where $\min(A)$ and $\max(A)$ form a range where it is possible to divide up this range into $2^n - 1$ buckets. The imaginary "zero" $\mathbb{O}$ position is the number of buckets it takes for $min(A)$ to each absolute 0. Hence, $f_{\text{GPTQ}_n(x)}^q = \big(\text{clamp}(\lfloor x/scale \rceil + \mathbb{O}, 0, 2^n - 1) - \mathbb{O}\big) \times scale$. Lastly, Rastegari et al. (2016) uses *XNOR* quantize function defined as $f_{\text{XNOR}(x)}^q = \text{sign}(x - \overline{A}) \times \text{scale} + \overline{A}, \forall x \in A$, where $scale = \overline{|x - \overline{A}|}$.

It is beneficial to process $W$ of size $[k, d]$ in group chunk $g \ll d$ (typically $g = 128$). The reason is to have a more localized "mean" and scale that resemble the group rather than resemble the whole row. Therefore, it can improve the quantization quality. This method is employed by many previous works like Huang et al. (2024); Frantar et al. (2022); Yuan et al. (2024); Yu et al. (2022). As later mention in the section 3.1, this costs more flag bits per row ($\lceil d/g \rceil$ more flags per row), but yield higher performance as previously shown in the literature. Error corrections for the subsequent groups are calculated as in eq. (1). The "st" and "ed" in eq. (1) are start and end of the current group that the matrix are being quantized on, where "ed:" indicates the range of indices at or after "ed" (similar to Python annotation of array). "diag" is to get the diagonal of the Hessian.

$$W[:, ed :] = \big(W[:, st : ed] - W_q[:, st : ed]\big) \times diag(H[st : ed, st : ed]) \times H^{-1}[st : ed, ed :] \quad (1)$$

GPTQ quantizes in group ($g = 128$) and uses Hessian metric to conduct two folds of error correction. The first fold is "within-group": $G = [j_1, j_2, \ldots, j_g]$, the error on $j_m$ will be corrected for all $j_i$ where $i \in [m + 1, g)$. The second fold is 'between-group': for every group $G_0 = [0, g); G_1 = [g, 2g), \ldots, G_n = [g_n, d)$, where the error on $G_m$ will be corrected for all following group $G_i$ where $i \in [m + 1, \lceil d/g \rceil]$ before continuing to quantize. BiLLM and PB-LLM are built on top of GPTQ, but only correct "between-group" and not "within" group, (this scheme is denoted as "Matrix-No Group" scheme). In addition, different from GPTQ, PB-LLM and BiLLM make a certain percentage of the groups (treatment groups) to become higher precision. PB-LLM uses the salient score to decide on which the treatment groups are, while Bi-LLM uses "High Order Residual" scheme which chooses based on not only salience but also the bell-shape distribution of the weights. As a result, we uses parameters to refers to those works, as in PB-LLM(8,1,0.1) and BiLLM(2,2,0.1). Refer to table 4 for meaning of those parameters. Secondly, PB-LLM and BiLLM only does "between" but not "within" group. *. Because error correction applies to what come after, it also has an ordering side-effect . We conduct a small experiment to confirm that we do need both corrections (Matrix-Group) for this work and the order of correction should be kept as default (no column sorting based on salience). Refer to table 8 and fig. 7 in appendix A.4. Lastly, there is kernel

---

[1] $\overline{A}$ is used for denoting "mean" value of $A$ in this paper.

that helps with the inference after the model being quantized like in LUT-GEMM (Park et al., 2022) or GPTQ Frantar et al. (2022). However, in terms of creating a GPU kernel that is for quantization, to the best of our knowledge, currently ours (k-OOC) is the first of its kind.

## 2 PROBLEM STATEMENTS

Hessian based quantization of a weight matrix $W$ in a linear module $A(X, W) = X \times W^T$ is to find a compressed version of $W$ called $W_q$ so that the loss function $L(W, W_q, X)$ is minimized (eq. (2)). $\min_{W_q}(L, W, W_q, X) = \overline{L}(W, W_q, X)$ has an approximate closed form in eq. (6), where each each row of its can be calculated according to eq. (3) and $H = 2X^T X$. See appendix A.2.1 for derivation.

$$L(W, W_q, X) = ||XW^T - X \times W_q^T||_2^2 \tag{2}$$

$$\overline{L_j} = \sum_{i=1}^{d} \left( H_{ii} \Delta W_{ji}^2 \right) \Rightarrow \overline{L} = \sum_{j}^{k} \left[ \sum_{i=1}^{d} \left( H_{ii}(W_{ji} - quant(W_{ji}))^2 \right) \right] \tag{3}$$

$\overline{L}(W, W_q, X)$ can be used as a predictor on $L(W, W_q, X')$ where $X'$ is a batch of unseen test points. Furthermore, previous literature described $\mathbb{S}(W, i, j) = H_{ii} \sum_j W_{ji}^2$ as the salient score of column $i$ of matrix $W$, and since $W_{ji} \neq \Delta W_{j,i}$, this salient metric is only used as a ad-hoc estimator of the $\overline{L_j}$. Secondly, schemes like GPTQ, PB-LLM, and BiLLM have different way to defining function $W_q = f^q(W)$. The first problem statement is to create a quantize function $f^q$ to minimize the error $L$ on a group of unseen $X$, using the knowledge of $\overline{L}$ and curvature of $L$ through $H$, under some quantize budget or Bit Family constraints (see section 3.1).

The quantizing problem expands to layer and model level, as module level quantized result cannot be used to predict the PPL of the whole model. A quantize process starts with a model $\mathbb{M}$, comprising of a list of layers $\mathbb{L}=\{\mathbb{L}^1, \mathbb{L}^2, \mathbb{L}^3,..\}$. Layer $\mathbb{L}^1$ comprises of a list of modules $\mathbb{L}^1_{\text{modules}}=\{\mathbb{L}^1_1, \mathbb{L}^1_2, \mathbb{L}^1_3,...\}$ and so on. Only linear modules are considered in this process. The "quantize train input" $\mathbb{L}^1_{\text{input}}=X$ is first feed into $\mathbb{L}_1$ to capture the inputs $\mathbb{L}^1_{\text{modules inputs}}$ to each of $\mathbb{L}^1_{\text{modules}}$. Each of those modules are then quantized independently with $\mathbb{L}^1_{\text{module inputs}}$ using $f^q_{\text{GPTQ}(l)}$. After quantizing, the input X is feed through the layer again (now with new weights) create the new output $\mathbb{L}^1_{\text{output}}$. $\mathbb{L}^2_{\text{inputs}}=\mathbb{L}^1_{\text{output}}$, and the process continues until it reaches the last layer. However, when $f^q_{\text{GPTQ}(l)}$ is replaced with $f^q_{\text{new}}$, and $f^q_{\text{new}}$ yields smaller loss for all modules than $f^q_{\text{GPTQ}(l)}$, the final module $\mathbb{M}(f^q_{\text{new}})$ might not have better PPL compared to $\mathbb{M}(f^q_{\text{GPTQ}(l)})$. Such relationship of module-wise and layer/model-wise quantization has not been fully explored in past literature. Method to solve this accurately and efficiently without compromising the quantize speed is discussed in section 3.3.

## 3 METHODOLOGY

### 3.1 BIT FAMILY: THE EFFECTIVE NUMBER OF COMPRESSION BITS

The aforementioned $f^q_{\text{sign}}$ method uses 1 flag to capture the "scale", because the "mean" is always fixed at 0. By the same token, $f^q_{XNOR}$ (in PB-LLM), uses 2 sets of {mean,scale}'s, namely {mean$_h$, scale$_h$} and {mean$_l$, scale$_l$}, to apply on different fragments of $W$. {mean$_h$, scale$_h$} is used for high precision (quantized with high # of bits) and {mean$_l$, scale$_l$} for lower precision (low # of bits). Hence, this scheme uses 4 flags. Lastly, BiLLM uses 3 sets of {mean,scale}'s, hence comprising 6 flags. Each of the flag is typically a Half float number (16 bits), thus the total number of *flag bit count* for BiLLM is $6 \times 16 = 96$ bits. On the other hand, the *scale* in $f^q_{GPTQ(b)}$ is a Half float, but the marking of imaginary zero $\mathbb{O}$ uses the same number of bits as $b$, hence the total number of flag bit count is $b + 16$.

When calculating the final effective bits of a quantized post-training algorithm, apart from the *flags bit count*, *mark bit count* (referred to as "index storing" bits in (Yu et al., 2022)) should be taken into consideration. In PB-LLM and BiLLM schemes, the high and low precision are applying on different section of the array, hence it is required to mark the array of which portion is high/low

precision. For matrix $W$ of size $[k,d]$, it requires $kd$ mark bits. *High Order Residual* scheme in BiLLM quantizes 10% of $W$ with high precision; the rest 90% is further split into 2 ranges based on salience: 45% is for the lower salience and the other 45% for the higher. In total, it requires $100\% \times kd \times 1 + 90\% \times kd \times 1 = 1.9 \times n$ mark bits. The average bit count $A$ is defined as [quantization bit + flag bit count + mark bit count]/(# elements), and $A^{max}$ and $A^{lim}$ together defines a *bit family*. The appendix A.1 summarizes the details of the notations used in calculating the *bit family*.

For instance, for $g = 128$, $f = 16$, table 4 proves that bit family $A^{\lim}_{\text{BiLLM}(2,2,0.1)}=(2-0.1)+(6\times 16)/128+0.1\times 2+0.9=3.75$ and $A^{\lim}_{\text{PB-LLM}(8,1,0.1)}=2.75$. However, Huang et al. (2024) reported that BiLLM(2,2,0.1) has 1.1 effective bit rate, because considered the mark/ flag bit count $F$ and $D$ separately. Yu et al. (2022)'s estimation of $A^{\lim}_{\text{PB-LLM}(8,1,0.1)}$ as 2.7 is close to 2.75, but missed the flags bit count $F$. On the other hand, $A^{\lim}_{\text{GPTQ}(2)}= \frac{2+16}{128}+2 \approx 2.1$ Hence, even when BiLLM(2,1,10%) has better PPL score than GPTQ(2), it is not objective to compare them because they are of different bit families. GPTQ(2) is a lot more compressed than BiLLM(2,2,0.1) and PB-LLM(8,1,0.1). For those reasons, we do **not** consider BiLLM(2,2,0.1) or PB-LLM(8,1,0.1) SOTA for $A^{lim} \leq 2.1$ family.

### 3.2 GPTQ-OOC: FINDING THE ODD ONE GROUP TO MAKE CLEARER PRECISION

Section 3.1 shows that the *mark bit count* affects the bit family (compression rate) tremendously for BiLLM and PB-LLM algorithm. Instead, it is beneficial to save storage by quantizing with $f^q_{GPTQ(l)}$, while incorporating the enhance higher (clearer) precision to a few columns in a selected group. The insight is that quantizing p portion of **one** group of $W$ into a higher precision with $f^q_{GPTQ(h)}$ (where the rest $1-p$ portion of that group and other groups are quantized in low resolution with $f^q_{GPTQ(l)}$) does **not** affect the bit family. Mark bit size M is unchanged as 0. The bit family of this proposed extension GPTQ(h,l,p) is in eq. (5). $h, l$ stands for the # of bits used for high/low precision.

$$F_{\text{odd}} = \frac{2(h+f)k}{kd}; F_{\text{others}} = \frac{(d/g-1)(l+f)k}{kd}; B = \frac{(pgh+(d-pg)l)k}{kd}$$

$$A = M + F_{\text{odd}} + F_{\text{others}} + B = \frac{l+f}{g} + l + \left[\frac{2(h+f)}{d} + \frac{pgh}{d} - \frac{l+f}{d} - \frac{lpg}{d}\right] \quad (4)$$

$$\Rightarrow A^{lim}_{GPTQ(h,l,p)} = \lim_{d\to\infty} A = \frac{l+f}{g} + l = A^{lim}_{GPTQ(l)} \quad (5)$$

See derivation of eq. (4) in appendix A.2.3. Equation (5) shows the core idea of GPTQ(h,l,p) that it maintains the same bit family as GPTQ(l) even with the introduction of a $h$ (high) precision. The one caveat is that the last few components of $A$ in eq. (4) can degrade when $d$ is small. OPT-125M has $d_{min} = 768 \Rightarrow A_{max} = A_{d=768} = 2.16$. Figure 1 shows changes of $A$ with respect to $d$ and $h$ when keeping the other parameters constant (p=0.1, $l = 2$, and g=128). To keep it in the same bucket as $A^{max}_{\text{GPTQ}(2)}= 2.1$, the chart suggests to pick $h = 2$ to maintain the worst case of bit family $A^{max}_{\text{GPTQ}(2,2,0.1)}= 2.16$ and theoretical limit bit family of $A^{\lim}_{\text{GPTQ}(2,2,0.1)}= 2.1$. Picking an odd group out of $\lceil d/g \rceil$ groups is not a trivial problem due to run-time constraint (PTQ method should be faster than QAT, ideally less than 4 hours from previous benchmarks). The solution is discussed in section 3.3.

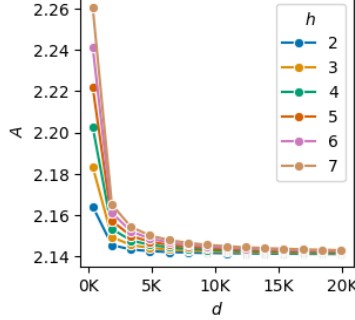

Figure 1: Changes in $A^{\max}_{\text{OOC}(h,2,10\%)}$ with respect to $d$ and $h$. The smallest value of $d = 768$ corresponds to the smallest model OPT-125M tested in this work.

### 3.3 ROW-FLOW-SELECTION PARALLEL, INPUT BATCHING , AND SPECIALIZED GPU FUSED KERNEL FOR OOC

Equation (3) shows that each row $j$ of $\overline{L}$ can be calculated independently of another row. This is also true for any quantize function $f^q \in [f^q_{\text{GPTQ}}, f^q_{\text{PB-LLM}}, f^q_{\text{BiLLM}}]$ (**Row Parallel**). A

CUDA device with capability 9 can execute 32 blocks concurrently for each multiprocessor (SM) (nvi). For instance, NVIDIA H100 has 132 SMs, hence can compute $\mathbb{P}_{\text{block}}$=132x32=4,224 units (rows) simultaneously. An information needed to compute $f^q(W_j^{\text{gid}})$ where $gid$ is the group id is the original weight $W$, diagonal of the Hessian $diag(H)$, $H^{-1}$ calculated by the fast Choslesky decomposition (Krishnamoorthy & Menon, 2013), and the pointer to the result and error matrix $W_q$ and $E$. There is another level of parallelism is to use maximum number of threads per SM (2048 in the case of H100), which can leads to $\mathbb{P}_{\text{thread}}$=132x2048= 270,336 units (rows) calculated at once. Only the block parallelism (1 thread per block) is considered in this work due to SRAM cache size limit (49KB) per blocks (see explanation in appendix A.3). Secondly, only quantizing of rows of $W$ is parallelized, quantizing different groups (in $\lceil d/g \rceil$ groups) cannot be parallelized, due to their sequential dependency. eq. (1) shows that later groups are depending on earlier group for error correction.

$f_{\text{GPTQ(h},l,\text{p})}^{\text{q}}$ only operates on **one** selected group (called clear group as quantized with higher # of bits), where the rest of the groups are quantized with $f_{\text{GPTQ}(l)}^{\text{q}}$. To pick the the best selection of an odd clear group, a brute-force method can be employed. However, it is also possible to use **selection parallel** similar to the row parallel. Yet, even with such parallelism, it is not practical to scan through all the groups due to time cost, but to focus on certain groups. A scan ratio $s$, where $c_{max} = s\lceil d/g \rceil$ groups are scanned (brute-forced) to determine whether upgrading the group to clearer precision yields lower error $L$. When $s < 1$, picking which groups to scan can be based on the salient metric of the group, $\mathbb{S}_{\sum}(W, \text{group\_id}) = \sum_{i \in \text{group\_id}} \mathbb{S}(W, i)$, for all $\text{group\_id} \in [0, \lceil d/g \rceil]$, etc. picking top $s\lceil d/g \rceil$ groups with $\mathbb{S}_{\sum}(W)$ sorted in descending order. All $c_{max}$ matrices of $W_q$ is audited against a "probe" input point (can be the last item of $X$ to save bandwidth), etc. picking the $W_q$ that yields the lowest $L(W_q, X_{probe})$ according to eq. (6). Therefore, $s$ should be added as the fourth hyper parameters, as in GPTQ(h,$l$,p,s).

OOC(h,$l$,p,s) is defined as a quantize scheme where it picks the best model out of one created from $f_{\text{GPTQ}(l)}^{\text{q}}$ and from the extension $f_{\text{GPTQ(h},l,\text{p,s})}^{\text{q}}$ using some validation dataset. In order to achieve that, it needs to run quantize process twice, each with different $f^q$ and sets of $\mathbb{L}_{\text{input}}^{\text{i}}$ and $\mathbb{L}_{\text{output}}^{\text{i}}$ flowing through the process. Hence, we define each run as a workflow. Figure 2a describes OOC scheme visually. The generic quantize function becomes $f_{\text{combine}}^{\text{q}}$: $(W, [X_{f_1}, X_{f_2}]) \rightarrow [W_{\text{q}}^{f_1}, W_{\text{q}}^{f2}]$, where $f_1, f_2$ are different quantize functions. For the OOC scheme, it is $f_{\text{OOC}}^{\text{q}}$:$(W, [X_{\text{GPTQ}(l)}, X_{\text{GPTQ(h},l,\text{p,s})}]) \rightarrow [W_{\text{q}}^{\text{GPTQ}(l)}, W_{\text{q}}^{\text{GPTQ(h},l,\text{p,s})}]$. Theoretically, this double workflow can be extended into triple or quadruple workflow, but the memory consumption of storing the input and different $W_q$ needs to be taken into consideration (Refer to section 3.4). From the perspective of block-parallelism aforementioned, double workflow can also be parallelized (**flow parallel**), by sharing aforementioned $\mathbb{P}_{\text{block}}$ units.

The sequential (no row-flow-selection parallel) version of OOC scheme p-OOC-Naive is described in Algorithm appendix A.3 ("p" stands for Python to indicate non-kernel fashion, as kernel is done in C++). p-OOC-Naive has 4 nested for-loops, where 3 of them (in highlight) can be avoided using the row-flow-selection parallel k-OOC in Algorithm algorithm 2. For a fairer comparison, we also create p-OOC-Batch variation where we only incorporate Selection Parallel, and not Row or Flow Parallel. Differing from Flow Parallel, where the result $W_q$ of each flow (referred to as "artifacts") are kept (first on GPU, then offloaded to CPU to save GPU space) during the whole *model* quantize process, Selection Parallel results are discarded after the *module* being quantized. Another implementation detail is that it is not possible to store all artifacts of $c_{max}$ groups in GPU memory at once, as for large model (up to 70B), a small ratio $s$ can lead to Out-Of-Memory. We derive a formula to calculate a smaller chunk $c < c_{max}$ of those groups to be scanned in section 3.4. Figure 2b visualizes the data-flow of this specialized fused kernel **k-OOC**. For this GPU kernel to launch, all inputs and outputs have to already have allocated spots in memory. For Flow Parallel, artifacts include: 1) inputs of each flow $X_{\text{GPTQ}(l)}$ and $X_{\text{GPTQ(h},l,\text{p})}$ (each has its derivatives like $H^{-1}$, diag(H)), and 2) outputs of each flows ($W_q, E$). This explains why $H^{-1}, E, W_q$ have first dimension of $n = 2$ (numbers of flows) in the figure. For Selection Parallel, $c$ selection artifacts need to be allocated, which explains the next dimension of $W_q$ and $E$ is $c$. Each selection and flow uses the same $W$, hence storing $W[n, c, k, d]$ is not necessary. Input $X$ to each selection per flow is the same, rendering the Hessian inverse $H^{-1}$ the same and no dimension of $c$ in $H^{-1}$. Diagonal of $H$ is not

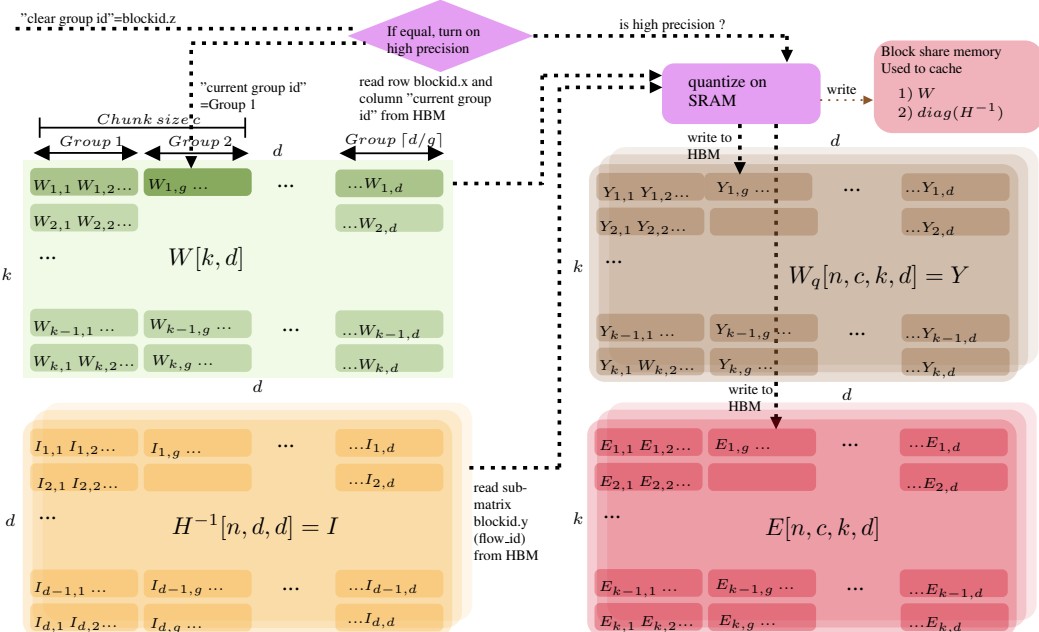

(a) Two independent workflows with their own inputs through the set of layers. The input is first passed to each module (the "forward" method) to capture the inputs for quantization and then forwarded again during "retest" to capture the new output to pass to the next layer as input.

(b) Specialized GPU kernel for the OOC algorithm where a combination of [row, flow, clear group id selection] are processed in parallel, which leads to improved quantization speed. $n$ stands for the number of workflows. $c$ stands for the number of "clear group id" to scanned through at once. Thick arrow indicates the read/write from/to high-latency HBM storage and thin arrow indicate low-latency read/write from/to block share cache memory.

Figure 2: The core pillars of k-OOC technique is the creation of double workflows in fig. 2a and batching those workflows inputs for parallel processing in a novel GPU kernel in fig. 2b).

shown in the figure; by the same logic, its dimension is $[n, d]$. This artifacts creation is called **Input Batching**.

## 3.4 K-OOC GPU MEMORY CONSIDERATION AND CHUNK SIZE CALCULATION

It is required to store the new quantization the artifacts of Flow and Selection Parallel ($W_q$, $E$, $H^{-1}$, and $H_{diag}$) at all time to maximize parallelism, a careful handling of GPU memory is necessary. Recommended steps are 1) loading model all weights on CPU and 2) only load to GPU the weight of the current layers being quantized. After quantization, those new weights ($W_q$) should be offload to CPU again to save space. When quantizing small models, one GPU can handle all tasks: 1) load the input, 2) do the first forward pass using $W$ to capture the input to each module, 3) calculate the inverse hessian, 4) quantize the model, 5) do the second forward pass to recalculate the new output using $W_q$ (fig. 2a). However, for large models (more than 3B parameters), step 4) should be offloaded to another GPU. Secondly, the formula to find a $c < c_{max}$ that the artifacts of $c$ selections is $c = \frac{T/f - [kd + n(md + mk + kd + d + d^2)]}{n(1 + kd + 2mk + 2kd)}$. The quantities in the equation are explained in the same order

in table 5. $f = 4$ is the bytes size of each item in those matrices. $T$ is the total GPU HBM memory in bytes (etc. 80GB for H100).

## 4 EXPERIMENTS

### 4.1 DATASET AND EXPERIMENT SETUPS

As we want to be comparable to bit family of $A_{\text{GPTQ(2)}}^{\lim} = A_{\text{GPTQ(2)}}^{\max} = 2.1$, $f_{\text{OOC(2,2,0.1,s)}}^{\text{q}}$ is experimented using different values of $s \in [0.1, 0.2, 0.5, 1]$. Choices of $s$ does not affect $A^{max}$ or $A^{lim}$, but affects PPL and quantize time cost. Specifically, OOC(2,2,0.1,0.1) means it is an ensemble double-flow scheme that combine the result of GPTQ(2) and GPTQ(2,2,0.1,0.1). For brevity, OOC(2,2,0.1,s) is referred to as OOC(s) in the report. Quantizing happens on 2 GPUs NVIDIA H100 80GB HBM3. Due to GPU memory constraints, we cannot experiment with triple or quadruple workflow. Secondly, 3 datasets C4, Wikitext2, and PTB are utilized to measure PPL. To keep conformity, 256 data samples (each with 2048 tokens) from each dataset is used. To create quantized Model $\mathbb{M}$(C4), this work uses 50% of $X_{C4}$ dataset as quantize input set [2], 10% as Validation set, and 40% as Test set. 14 models experimented are of 3 family types: OPT, LLaMA, and BLOOM, ranging from 125M to 70B parameters. Any size up to 3B is considered "small", while up to 70B is considered "large" (fig. 3 and fig. 5). We also use 40% of the $X_{\text{Wikitext2}}$ and of $X_{\text{PTB}}$ to evaluate $\mathbb{M}$(C4) (table 1). We also create and report PPL of $\mathbb{M}$(Wikitext2) and $\mathbb{M}$(PTB) (table 7 and table 6). Thirdly, for speed measuring, we compare the sole effect of **Row Parallel**, by comparing the original GPTQ implementation with k-OOC(0). It is justified comparison because the workload of OOC(0)=OOC(2,2,0.1,0)=ensemble(GPTQ(2), GPTQ(2,2,0.1,0))=GPTQ(2) as GPTQ(2,2,0.1,0)=GPTQ(2) (we do not convert 10% of any groups into clearer precision). For **Row-Flow-Selection** combined parallelism, we compare k-OOC(s) with p-OOC-Naive(s) and p-OOC-Batch(s) (table 2). Lastly, memory consumption of k-OOC(s) for different $s$ (including $s = 0$) is tested and reported in fig. 6.

### 4.2 RESULTS

We summarize all PPL score comparison in table 3. The average improvement on PPL for all small models are bigger than for large models (8.9% vs 4.1%), and it is expected as large models are more tuned and have good PPL to begin with. The only exception is for $\mathbb{M}_{\text{Wikitext2}}$ where it has better PPL on large v.s. small models (thanks to the s=0.5, see table 6 [3]). It is expected that the average PPL improvement on test set is lower than validation set (8.9% v.s. 9.7%), as validation set compares $\min(\text{PPL}(f_{\text{GPTQ(2)}}^{\text{q}}, \text{Val set}), \text{PPL}(f_{\text{GPTQ(2,2,0.1,s)}}^{\text{q}}, \text{Val set}))$ with $\text{PPL}(f_{\text{GPTQ(2)}}^{\text{q}}, \text{Val set})$. With $f_{\text{OOC(s)}}^{\text{q}}$=ensemble($f_{\text{GPTQ(2)}}^{\text{q}}$, $f_{\text{GPTQ(2,2,0.1,s)}}^{\text{q}}$), test set compares $\text{PPL}(f_{\text{OOC(s)}}^{\text{q}}, \text{Test set})$ with $\text{PPL}(f_{\text{GPTQ(2)}}^{\text{q}}, \text{Test set})$. It is likely that $f_{\text{OOC(s)}}^{\text{q}}$ performs better than $f_{\text{GPTQ(2)}}^{\text{q}}$ on test set, but not always as shown in table 1 toward bottom of the table ("PTB T" and "WIK T" rows). Also in table 1, BiLLM(2,2,0.1) performs better than GPTQ(2) and k-OOC(s) but as $A_{\text{BiLLM(2,2,0.1)}}^{\lim}=3.75 > A_{\text{OOC(2,2,0.1,s)}}^{\lim}=2.1$, the comparison is not justified.

Quantization speed is an important metric in judging quantization algorithm quality. Figure 4 shows that k-OOC almost always performs faster than p-OOC-Naive and p-OOC-Batch, especially by a big margin for $s \in [0.0, 0.1, 0.2]$ and by less margin for $s \in [0.5, 0.1]$. Table 2 quantifies this gap numerically: in the single-flow (s=0.0), it shows that k-OOC improves up to 9x/4x the speed of GPTQ for small/large models just using the Row Parallel technique alone. In the double-flow, k-OOC can gain up to 22x speed up for small model, but the gain diminishes for large model. This is expected as the chunk-size degrades to $\sim 1$ when the dimensions of $W$ is big (see fig. 8). Finally, fig. 6 shows that k-OOC uses more memory than GPTQ, especially in the $s > 0$ cases.

---

[2]Post-Training Quantization process does not need as many data as regular training or fine-tuning

[3]Marker "-" indicates the test is not run for that case. "T" stands for "test" and "V" stands for "validation". For instance, "WIK T" means "Wikitext2 Test set'. See table 6 and table 7 for results of training on *Wikitext2* and *PTB*

| Method | Bit Fam | Eval on | OPT 125M | OPT 350M | BLOOM 560M | OPT 1.3B | BLOOM 1.7B | OPT 2.7B | BLOOM 3B | OPT 6.7B | LLAMA 7B | BLOOM 7B1 | OPT 13B | LLAMA 13B | OPT 30B | LLAMA 70B |
|---|---|---|---|---|---|---|---|---|---|---|---|---|---|---|---|---|
| GPTQ | 2.1 | C4 V | 148.50 | 152.33 | 76.49 | 43.40 | 44.12 | 35.04 | 27.08 | 16.31 | 30.34 | 19.09 | 14.08 | 13.61 | 10.73 | **9.53** |
| k-OOC(0.1) | 2.1 | C4 V | 135.96 | 119.21 | **65.03** | 43.40 | 40.49 | 30.44 | **26.16** | 16.11 | **26.43** | 19.09 | 13.94 | 13.45 | 10.70 | **9.53** |
| k-OOC(0.2) | 2.1 | C4 V | 125.47 | 149.31 | 68.71 | 43.40 | **37.86** | 30.96 | 26.97 | **15.81** | 28.12 | 19.09 | 13.94 | 13.61 | **10.68** | **9.53** |
| k-OOC(0.5) | 2.1 | C4 V | 133.05 | **109.38** | 66.25 | **42.16** | 43.26 | **30.43** | 26.82 | 16.05 | 30.34 | **18.91** | **13.89** | 13.33 | - | - |
| k-OOC(1.0) | 2.1 | C4 V | **122.19** | 130.70 | 67.20 | 43.40 | 41.42 | 32.28 | 26.47 | 16.16 | 27.36 | 19.00 | 14.00 | **13.27** | - | - |
| FP16 | 16 | C4 T | 22.14 | 18.73 | 21.70 | 13.20 | 15.98 | 11.82 | 14.32 | 10.51 | 6.00 | 12.42 | 9.87 | 5.53 | 9.26 | 4.58 |
| GPTQ | 2.1 | C4 T | 174.25 | 181.01 | 82.06 | 49.64 | 46.27 | 37.94 | 28.29 | 17.38 | 24.22 | 19.15 | 14.89 | 12.50 | 11.51 | **8.52** |
| k-OOC(0.1) | 2.1 | C4 T | 163.14 | 141.95 | **69.74** | 49.64 | 42.67 | 33.04 | **27.30** | 17.08 | **21.95** | 19.15 | 14.69 | **12.14** | 11.49 | **8.52** |
| k-OOC(0.2) | 2.1 | C4 T | 146.22 | 175.56 | 71.50 | 49.64 | **39.35** | 33.04 | 27.79 | **16.77** | 23.76 | 19.15 | 14.71 | 12.50 | **11.49** | **8.52** |
| k-OOC(0.5) | 2.1 | C4 T | 163.44 | **133.88** | 72.43 | **47.61** | 46.20 | **32.53** | 27.52 | 17.04 | 24.22 | **18.83** | **14.59** | 12.18 | - | - |
| k-OOC(1.0) | 2.1 | C4 T | **143.74** | 152.22 | 71.21 | 49.64 | 43.76 | 34.88 | 27.40 | 17.03 | 22.95 | 19.10 | 14.73 | 12.17 | - | - |
| FP16 | 16 | PTB T | 39.66 | 31.70 | 44.48 | 20.39 | 30.52 | 18.02 | 25.76 | 15.79 | 38.10 | 21.22 | 14.56 | 51.12 | 14.05 | 24.16 |
| GPTQ | 2.1 | PTB T | 622.33 | 752.43 | 376.85 | 157.54 | 162.31 | 107.34 | 87.67 | 31.14 | **7549.46** | 42.32 | 28.53 | 406.20 | 19.67 | 47.98 |
| k-OOC(0.1) | 2.1 | PTB T | 539.55 | **397.34** | 279.29 | 157.54 | 180.50 | 83.93 | 84.37 | 30.24 | 11053.28 | 42.32 | 27.21 | 374.43 | 19.65 | 47.98 |
| k-OOC(0.2) | 2.1 | PTB T | 477.50 | 772.19 | **268.88** | 157.54 | **144.88** | 90.25 | 89.51 | **28.35** | 15142.11 | 42.32 | 26.92 | 406.20 | **19.40** | 47.98 |
| k-OOC(0.5) | 2.1 | PTB T | 589.36 | 461.88 | 280.91 | **130.42** | 158.73 | **81.47** | **83.46** | 30.00 | 7549.46 | 42.83 | 28.21 | 420.46 | - | - |
| k-OOC(1.0) | 2.1 | PTB T | **423.89** | 487.73 | 329.00 | 157.54 | 171.44 | 91.86 | 86.47 | 29.31 | 18848.80 | 42.76 | 27.84 | **335.47** | - | - |
| FP16 | 16 | WIK T | 27.89 | 22.12 | 23.26 | 14.77 | 15.84 | 12.57 | 13.88 | 10.93 | 5.65 | 11.70 | 10.21 | 5.05 | 9.60 | 3.48 |
| PB-LLM(*) | 2.75 | WIK T | - | - | - | 265.52 | - | 124.35 | - | 105.16 | 69.20 | - | 81.92 | 151.09 | 25.14 | 28.37 |
| BiLLM(**) | 3.75 | WIK T | - | - | - | 69.97 | - | 49.55 | - | 35.36 | 32.48 | - | 18.82 | 16.77 | 12.71 | 8.41 |
| GPTQ | 2.1 | WIK T | 378.49 | 508.79 | 131.32 | 101.14 | 68.38 | 74.95 | 34.75 | 21.54 | 46.49 | 20.26 | 23.01 | 15.21 | 13.25 | **9.09** |
| k-OOC(0.1) | 2.1 | WIK T | 347.73 | 319.90 | **114.24** | 101.14 | 59.49 | 58.81 | 33.80 | **20.81** | **36.66** | 20.26 | 21.73 | 15.73 | **13.20** | **9.09** |
| k-OOC(0.2) | 2.1 | WIK T | 303.53 | 480.37 | 116.72 | 101.14 | **53.59** | 61.66 | 32.31 | 20.89 | 114.08 | 20.26 | 22.04 | 15.21 | 13.39 | **9.09** |
| k-OOC(0.5) | 2.1 | WIK T | 513.89 | **297.54** | 117.65 | **83.48** | 64.26 | **57.68** | 33.43 | 20.93 | 46.49 | 20.19 | **21.01** | 15.47 | - | - |
| k-OOC(1.0) | 2.1 | WIK T | **300.29** | 375.04 | 116.91 | 101.14 | 61.17 | 70.19 | **31.78** | 21.13 | 38.31 | **20.13** | 22.12 | **14.89** | - | - |

Table 1: Perplexity (PPL) of **k-OOC (Ours)** compared against GPTQ(2) and Full Precision (Float 16). The lower the PPL, the better the model. It is quantized-trained on the *C4* dataset and tested on all 3 datasets. k-OOC(0.1) is short for k-OOC(2,2,0.1,0.1) . The table shows that all variations of k-OOC at least out-performs the GPTQ(2). The best performance compared against GPTQ(2) among different "Scan" percentages is in **bold**. (*) and (**) are reports from (Huang et al., 2024) and (Yuan et al., 2024) for PB-LLM(8,1,0.1) and BiLLM(2,2,0.1) for reference.

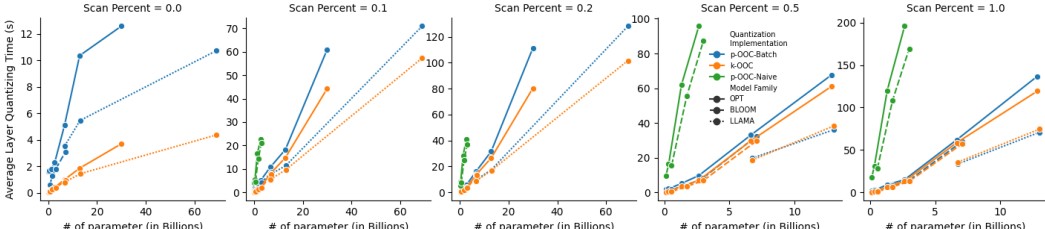

Figure 4: Average time (in secs) taken to quantize a **layer** sliced by different model (size and family) and different implementations of the OOC scheme. The speed is measured on H100s. It shows that the kernel version **k-OOC** costs less time compared to *p-OOC-Naive* and *p-OOC-Batch*. When the *Scan percent=0.0*, the p-OOC-Naive / p-OOC-Batch becomes the original GPTQ implementation in (Frantar et al., 2022). There is no difference between the Naive and Batch in this case, because the # of work flow is both 1.)

## 5 CONCLUSION AND LIMITATION

In this work, we developed the first world specialized fused GPU kernel for PTQ process, where it can reach $9\times$ faster than the original $f^{\mathrm{q}}_{\mathrm{GPTQ}(l)}$ implementation due to the usage of Row Parallel technique. Secondly, we introduced an extension to GPTQ($l$) called $f^{\mathrm{q}}_{\mathrm{GPTQ}(\mathrm{h},l,\mathrm{p,s})}$, where it upgrades

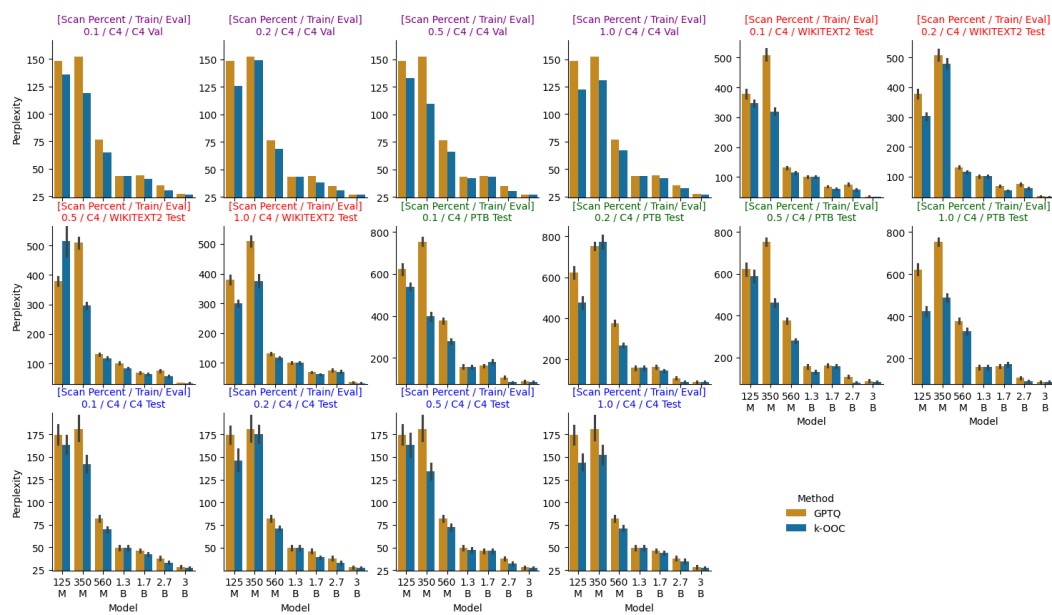

Figure 3: Perplexity of small models (up to 3B parameters) sliced by $s$, Train Dataset, and Eval Dataset, and Methods. See fig. 5 for PPL of bigger models.

| Quantization Implementation | Small Models (Up to 3B) / Scan Percent | | | | | Large Models (Up to 70B) / Scan Percent | | | | |
|---|---|---|---|---|---|---|---|---|---|---|
| | 0.0 | 0.1 | 0.2 | 0.5 | 1.0 | 0.0 | 0.1 | 0.2 | 0.5 | 1.0 |
| p-OOC-Naive | x1.00 | x1.00 | x1.00 | x1.00 | x1.00 | - | - | - | - | - |
| p-OOC-Batch | x1.00 | x4.07 | x6.26 | x9.62 | x12.49 | x1.00 | x1.00 | x1.00 | x1.00 | x1.00 |
| **k-OOC** | **x9.86 (Faster)** | **x16.24** | **x17.68** | **x20.90** | **x22.84** | **x4.01** | **x1.27** | **x1.18** | **x1.06** | **x1.03** |

Table 2: Speed up of **k-OOC** (algorithm 2) to the two Python implementations (p-OOC-Batch and p-OOC-Naive).

| Scan Percent | C4 Val | | C4 Test | | PTB Val | | PTB Test | | WIKITEXT2 Val | | WIKITEXT2 Test | | Validation | | Test | |
|---|---|---|---|---|---|---|---|---|---|---|---|---|---|---|---|---|
| | Small | Large | Small | Large | Small | Large | Small | Large | Small | Large | Small | Large | Small | Large | Small | Large |
| 0.1 | 10.0% | 2.4% | 9.6% | 2.2% | 9.2% | 0.9% | 6.4% | 1.0% | 6.9% | 1.0% | 5.1% | 0.8% | 8.7% | 1.4% | 7.1% | 1.3% |
| 0.2 | 7.7% | 1.7% | 8.8% | 1.0% | 13.2% | 1.9% | 11.7% | 1.6% | 7.3% | 20.5% | 5.5% | 16.4% | 9.4% | 8.0% | 8.7% | 6.3% |
| 0.5 | 10.1% | 1.2% | 9.3% | 1.6% | 5.5% | 2.3% | 6.3% | 2.3% | 11.1% | 18.0% | 10.0% | 18.0% | 8.9% | 7.2% | 8.5% | 7.3% |
| 1.0 | 8.6% | 2.9% | 9.0% | 2.2% | 13.5% | 1.4% | 12.3% | 1.2% | 13.3% | 1.2% | 12.3% | 1.3% | 11.8% | 1.8% | 11.2% | 1.6% |
| Average | 9.1% | 2.0% | 9.2% | 1.8% | 10.4% | 1.6% | 9.2% | 1.5% | 9.7% | 10.2% | 8.2% | 9.1% | **9.7%** | **4.6%** | **8.9%** | **4.1%** |

Table 3: Final PPL improvement of k-OOC(s) compare to the GPTQ baseline in terms of percentage. "Small/ Large" stands for the size of the models (Up to 3B is consider "small"). Scan percentage tested are s=$\{0.1, 0.2, 0.5, 1\}$.

**one** group into higher precision by using two bit sizes $h$ and $l$. However, due to error correction and model-wise aggregation, $f^{\mathrm{q}}_{\mathrm{GPTQ}(h,l,p,s)}$ can sometimes degrade compare to $f^{\mathrm{q}}_{\mathrm{GPTQ}(l)}$, we ensemble the two to create the final OOC(h,$l$,p,s) and incorporate row-flow-selection parallel into the earlier GPU kernel to improve speed. Empirically, we show that OOC(2,2,0.1,s) performs better than GPTQ(2) by 8.9% on small models and 4.1% on big models while still maintaining a good speed. We managed to quantize the big LLaMA 70B with $s \in [0.1, 0.2]$ under 2.5 hours. A limitation of this work is that the used parallelism scheme is GPU block parallelism where thread parallelism can further improve the speed. However, the true bottle neck lies in the memory consumption of OOC, where a technique of not materializing $W_q$'s and $E$'s for all selections at once on GPU is needed. However, $E$ is essential for the error correction process of Hessian-base PTQ, so it remains the hard question and will be a topic for future work.

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

| Algorithm | Description and Bit Family calculation |
|---|---|
| GPTQ(b) | $W$ is quantized at $b$ bits per item. 1 flag is used per group per row. Hence $F = \frac{(b+1f)\frac{d}{g}\times k}{kd} = \frac{b+1f}{g}$; $M = 0$; $B = \frac{bkg}{kd} = b \Rightarrow$ $\boxed{A^{lim} = \lim(F+B) = \frac{b+f}{g} + b = A_{max}}$ |
| PB-LLM(h,$l$,p) | The method combines p% high and (1-p)% low precision. For high precision, it uses the GPTQ(h) above but with 1 flag per row (not per group). For low precision, it uses XNOR$_l$ method (2 flags per group per row). It needs $kd$ bit to mark high/low precision. Hence, $F_{low} = \frac{2f(d/g)k}{kd} = \frac{2f}{g}$; $F_{high} = \frac{(h+1f)k}{kd} = \frac{h+1f}{d}$; $M = \frac{kd}{kd} = 1$; and $B = \frac{(hp+l(1-p))kd}{kd} = ph + (1-p)l \Rightarrow A^{lim} = \lim(M + F_{low} + F_{high} + B) = \boxed{1 + \frac{2f}{g} + \frac{h+f}{d} + ph + (1-p)l = A_{max}}$ |
| BiLLM(h,$l$,p) | The method further splits the low range into 2 parts. Hence it combines p% at h bits, (1-p)/2% for upper low at $l$ bits, and (1-p)/2% for lower low at $l$ bits. 4 flags per group per row is for high precision. For each range of low precision, 2 flags per group per row are used. Hence, there are 8 degrees of freedom (8 flags) in total. It requires $kd$ bit to mark high/low precision and another $pkd$ to mark upper/lower low precision. Hence, $M = \frac{kd+(1-p)kd}{kd} = 2 - p$; $F_{high} = \frac{hf}{g}$, $F_{low} = \frac{2lf}{g}$. $B = 0.1h + 0.9l \Rightarrow A^{lim} = \lim(M + F_{high} + F_{low} + B) = \boxed{(2-p) + \frac{(h+2l)f}{g} + ph + (1-p)l = A_{max}}$ |

Table 4: Bit Family calculation for 3 main algorithms

# A   APPENDIX

## A.1   BIT FAMILY NOTATION

**"Bit Family" notations**

| | |
|---|---|
| $k, d$ | The dimension of the weight matrix that needs to be quantized |
| $f$ | The number of bit that is used for a flag (Typically Half float $f = 16$ bits) |
| $g$ | The group size that is processed at one time which also defines the range at which quantization stats (mean, scale) are calculated on. $d/g$ is equals the number of groups per row |
| $h, l, b$ | The number of bits for "high", "low", and "regular" precision tiers that are used to quantize numbers in the GPTQ, PB-LLM, and BiLLM schemes |
| $p$ | The proportion within each group to quantize with high precision (typically $p = 10\%$) |
| $F(k, d)$ | Average *flag bit count*, etc. number of bits per slot to store the flags for $W$ of size $[k, d]$ |
| $M(k, d)$ | Average *mark bit count*, etc. number of bits per slot to mark which items are of high/low precision for $W$ of size $[k, d]$ |
| $B(k, d)$ | Average number of bits per slot to quantize $W$, not including $F$ and $D$ |
| $A^{max}$ | Max Average number of bits per slot. A=M+F+B $\rightarrow A_{max} = \max_{k,d}$(M+F+B) |
| $A^{lim}$ | Average Final Bit Family at limit. $A^{lim} = \lim A(k,d) = \lim_{w,d\to\infty} \big[$M(k,d)+F(k,d)+B(k,d)$\big]$ |

## A.2 EQUATION DERIVATIONS

### A.2.1 THE SALIENT METRIC $\overline{L}$

Using Taylor expansion on eq. (2), we have:

$$L(W, W_q, X) = L(W + \Delta W, X) = L(W) + ||\triangledown(W)\Delta W^T + \frac{1}{2}\Delta W \triangledown^2(W)\Delta W^T||_2^2 \quad (6)$$

In eq. (6), $\triangledown(W)$ is the Jacobian matrix of L with respect to $W$, and $\triangledown^2(W)$ is the Hessian matrix of L. The hessian has a closed form of $\triangledown^2(W) = 2X^T X$, independent of $W$. When $W_q = W$, $L(W, W_q, X) = L(W, W, X) = ||XW^T - XW^T||_2^2 = 0$, hence $L(W, W, X)$ is minimal, which leads to $\triangledown(W) = 0$. The first two terms of eq. (6) becomes zeros, and $\min(L)$ becomes:

$$\overline{L(W, X)} = \min_{W_q}\Big(L(W, W_q, X)\Big) = \min_{W_q}\Big(||\Delta W_q X^T X \Delta W_q^T||_2^2\Big) \quad (7)$$

An insight is that each row $j$ of $\bar{L}$ can be calculated **independently** from other rows.

### A.2.2 SALIENT SCORE AND $\bar{L}$

Previous literature simplified the relationship between non diagonal entries in the Hessian matrix to 0. From equation eq. (7):

$$\overline{L_j} = \begin{bmatrix} \Delta W_{j1} & \Delta W_{j2} & ... & \Delta W_{jd} \end{bmatrix} \begin{bmatrix} H_{11} & H_{12} & ... & H_{1d} \\ H_{21} & H_{22} & ... & H_{2d} \\ ... & ... & ... & ... \\ H_{d1} & H_{d2} & ... & H_{dd} \end{bmatrix} \begin{bmatrix} \Delta W_{j1} \\ \Delta W_{j2} \\ ... \\ \Delta W_{jd} \end{bmatrix}$$

$$\approx \begin{bmatrix} \Delta W_{j1} & \Delta W_{j2} & ... & \Delta W_{jd} \end{bmatrix} \begin{bmatrix} H_{11} & 0 & ... & 0 \\ 0 & H_{22} & ... & 0 \\ ... & ... & ... & ... \\ 0 & 0 & ... & H_{dd} \end{bmatrix} \begin{bmatrix} \Delta W_{j1} \\ \Delta W_{j2} \\ ... \\ \Delta W_{jd} \end{bmatrix} = \sum_{i=1}^{d}\big(H_{ii}\Delta W_{ji}^2\big)$$

$$\Rightarrow \overline{L} = \sum_{j}^{k}\Bigg[\sum_{i=1}^{d}\big(H_{ii}\Delta W_{ji}^2\big)\Bigg] = \sum_{j}^{k}\Bigg[\sum_{i=1}^{d}\big(H_{ii}(W_{ji} - quant(W_{ji}))^2\big)\Bigg]$$

### A.2.3 OOC AVERAGE NUMBER OF BITS PER SLOT $A(k, d)$

$$F_{\text{odd}} = \frac{2(h+f)k}{kd}; F_{\text{others}} = \frac{(d/g - 1)(l+f)k}{kd}; B = \frac{(pgh + (d - pg)l)k}{kd}$$

$$C = M + F_{\text{odd}} + F_{\text{others}} + B = \frac{2(h+f)}{d} + \frac{l+f}{g} - \frac{l+f}{d} + \frac{pgh}{d} + \frac{ld}{d} - \frac{lpg}{d}$$

$$= \frac{l+f}{g} + l + \Big[\frac{2(h+f)}{d} + \frac{pgh}{d} - \frac{l+f}{d} - \frac{lpg}{d}\Big]$$

### A.2.4 CHUNK SIZE $c$ CALCULATION

$$f \times (kd + nmd + nmk + nckd + nkd + nckd + nd + nd^2 + nckd + nc + ncmk + ncmk) = T$$

$$\Rightarrow nc(1 + kd + 2mk + 2kd) + kd + n(md + mk + kd + d + d^2) = T/f$$

$$\Rightarrow c = \frac{T/f - [kd + n(md + mk + kd + d + d^2)]}{n(1 + kd + 2mk + 2kd)} \quad (8)$$

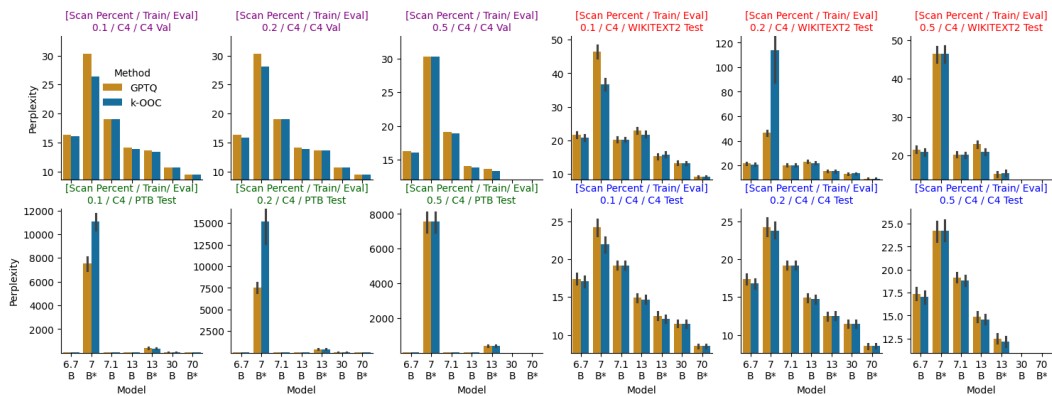

Figure 5: Perplexity of big models (up to 70B parameters) sliced by $s$, Train Dataset, and Eval Dataset, and Methods.

### A.3 OOC PSEUDO CODE AND IMPLEMENTATION DETAILS

There are two memory access scheme in GPU: low-latency Block shared memory (SRAM) and high-latency High Bandwidth Memory (HBM-DRAM) accesses. The inputs are originally loaded onto HBM, hence read/write to them is slow. Therefore, during calculating $f^q(W)$, the system is benefit from caching $W_j^{\text{gid}}$ into SRAM, because it will be read many times for calculation of the "mean" and "scale" in $f^q_{\text{GPTQ}}$ for example. In the detail implementation of GPTQ, there are about 9 lists (each with the size of $g = 128$ of 8 byte float numbers, should be cached. Hence, with the limit shared memory size per block of 49KB, only $\sim 5$ threads per block can be run. Compared to 1 thread per block (implemented in this paper), this extra threads per block might not yield much speed up, but worthy of exploration for future work.

| Matrices | Location in algorithm 2 | Dimensions |
|---|---|---|
| $W$ | 1 | $k \times d$ |
| Probe input | 2 | $n \times m \times d$ |
| Probe output before quant | 3 | $n \times m \times k$ |
| $W_q$ | 4 | $n \times c \times k \times d$ |
| Best $W_q$ | 5 | $n \times k \times d$ |
| E | 6 | $n \times c \times k \times d$ |
| Diagonal of $H$ | 7 | $n \times d$ |
| Inverse H | 8 | $n \times d \times d$ |
| Error correction | 9 | $n \times c \times k \times d$ |
| Norms of all clear group ids in chunk | 10 | $n \times c$ |
| Probe output after quant | 11 | $n \times c \times m \times k$ |
| Probe output differences | 12 | $n \times c \times m \times k$ |

Table 5: Dimensions of different quantities used in the k-OOC algorithm. This is to calculate the final "chunk size" $A$ so that the everything is fit in one GPU.

### A.4 EXTRA EXPERIMENTS

**Algorithm 1 OOC**: Quantizing a Matrix $W$ under OOC scheme using naive implementation. The 3 highlighted for-loops are the bottleneck of this algorithm.

---

    **function** GPTQ($W, diag, H^{-1}$,selected_group_id,g, high, low, $X_{\text{test}}$)
        k,d=W.shape
        $W_q = O_{[\text{k,d}]}$
        $E = O_{[\text{k,d}]}$
        **for** group_id $\leftarrow$ range(0, num_group) **do**
            start=group_id$\times g$
            end=(group_id+1)$\times g$
            **if** group_id=selected_group_id **then**
                precision$\leftarrow$ high
            **else**
                precision$\leftarrow$ low
            **end if**
            **for** row $j \leftarrow k$ **do**
                W$_q$[$j$, start:end], E[$j$ ,start:end]=quantize($W_j, diag, H^{-1}$,group_id, precision)
            **end for**
            W$_q$[:, end:] -= E[:,start:end]$\times H^{-1}$[ start:end, end:]                     ▷ Error correction
        **end for**
        **return** $W_q, L(W, W_q, X_{\text{test}})$      ▷ $L(W, W_q, X_{test})$ is the loss difference between W vs. $W_q$
    **end function**

    **function** OOC_INTERNAL($W, diag, H^{-1}, X_{\text{train}}$, scan_groups, g, high, low)
        $H^{-1}$=cholesky_inverse(H)
        min_loss$\leftarrow \infty$
        best$_{\text{W}_q}\leftarrow$ None
        **for** odd_group $\in$ scan_groups **do**
            $W_q$, loss= GPTQ($W, diag, H^{-1}$, odd_group, g, $X_{\text{train}}[-1]$)
            **if** loss$<$min_loss **then**
                min_loss$\leftarrow$ loss
                best$_{\text{W}_q}$=W$_q$
            **end if**
        **end for**
        **return** best$_{\text{W}_q}$
    **end function**

    **function** OOC($W$,work_flows, g, high, low)
        **for** work_flow $\in$ work_flows **do**
            $X_{\text{train}}$=work_flow['X']
            $s$=work_flow['s']
            $H = 2X_{\text{train}}^T \times X_{\text{train}}$
            // Calculate the salient metric $\mathbb{S}'$ per group.
            num_group=$\lceil d/g \rceil$
            S=$\big[\mathbb{S}'(W, group\_id)$ for group_id $\in [0, num\_group)\big]$
            S=sorted(S, descending=True)
            S=S[:,$s\times$num_group]               ▷ Only take the top s portion of the groups
            work_flow['result']=OOC_internal($W, diag(H), H^{-1}, X_{\text{train}}$, S, g, high, low)
        **end for**
        **return** work_flow
    **end function**

---

**Algorithm 2 k-OOC**: Quantizing a Matrix $W$ under OOC scheme using a specialized GPU kernel as described in section 3.3

---

**function** QUANTIZING_KERNEL($W, diag_{batch}, H^{-1}_{batch}, X_{probe\_batch}, S_{batch}$, g, high, low)

    row_id=block.idx                                    ▷ Called/ calculated on GPU2

    flow_id=block.idy

    option_id=block.idz

    scan_group=$X_{probe\_batch}$[flow_id][option_id]

    **if** group_id=scan_group **then**

        precision← high

    **else**

        precision← low

    **end if**

    quantize(row_id, $W, diag_{batch}, H^{-1}$,group_id, $W_q$[flow_id, option_id], E[flow_id, option_id], precision)

    **return** 0

**end function**

 

**function** OOC_KERNEL($W$,diag$_{batch}$,$H^{-1}$$_{batch}$, X$_{probe\_batch}$, S$_{batch}$, g, high, low)

    k,d=W.shape                                  ▷ Called on CPU, calculated on GPU 1

    num_flows=$S_{batch}$.shape[0]

    max_num_options=max($S_{batch}$.shape[1])

    $W_q = O_{[\text{num\_flows, max\_num\_options, k,d}]}$                           ▷ ④

    $E = O_{[\text{num\_flows, max\_num\_options, k,d}]}$                              ▷ ⑥

    **for** group_id ← range(0, num_group) **do**

        start=group_id×g

        end=(group_id+1)×g

        GPU Kernel call quantize_kernel($W, diag_{batch}, H^{-1}_{batch}$,group_id, $S_{batch}, W_q, E$)

        on grid of dimensions ≪k,num_flows, max_num_options≫

        $W_q$[:,:,:, end:] -= E[:,:,:,start:end]× $H^{-1}$[:, start:end, end:]      ▷ Error correction ⑨

    **end for**

    best$_{W_q}$=select$\left(W_q, L(W, W_q, X_{\text{probe\_batch}})\right)$                ▷ ③,⑤,⑩,⑪,⑫

    **return** best$_{W_q}$         ▷ $L(W, W_q, X_{probe\_batch})$ is the loss difference between W vs. $W_q$

**end function**

 

**function** OOC($W$,work_flows, g, high, low)         ▷ ①, called on CPU, calculated on GPU 1

    work_flow_map={}

    **for** flow_id, work_flow ∈ work_flows **do**         ▷ A fast For-loop for preparing the input.

        $X_{\text{train}}$=work_flow['X']

        $s$=work_flow['s']

        $H = 2X_{\text{train}}^T \times X_{\text{train}}$

        // Calculate the salient metric $\mathbb{S}'$ per group.

        num_group=$\lceil d/g \rceil$

        S=$\left[\mathbb{S}'(W, group\_id) \text{ for } group\_id \in [0, num\_group)\right]$

        S=sorted(S, descending=True)

        S=S[:,$s$×num_group]                 ▷ Only take the top s% of the groups

        work_flow_map[flow_id]={'H':H, 'probe': $X_{\text{train}}$[-1], 'S': S }

    **end for**

    $H^{-1}$$_{\text{batch}}$=[cholesky_inverse(h) for h in flatten(work_flow_map, 'H')]

    diag$_{\text{batch}}$=[diag(h) for h in flatten(work_flow_map, 'H')]              ▷ ⑦,⑧

    X$_{\text{probe\_batch}}$=flatten(work_flow_map, 'probe')                   ▷ ②

    S$_{\text{batch}}$=flatten(work_flow_map, 'S')

    result=OOC_internal(W, diag$_{\text{batch}}$,$H^{-1}$$_{\text{batch}}$, X$_{\text{probe\_batch}}$, S$_{\text{batch}}$, g, high, low)

    **return** map_result_to_work_flows(result)

**end function**

---

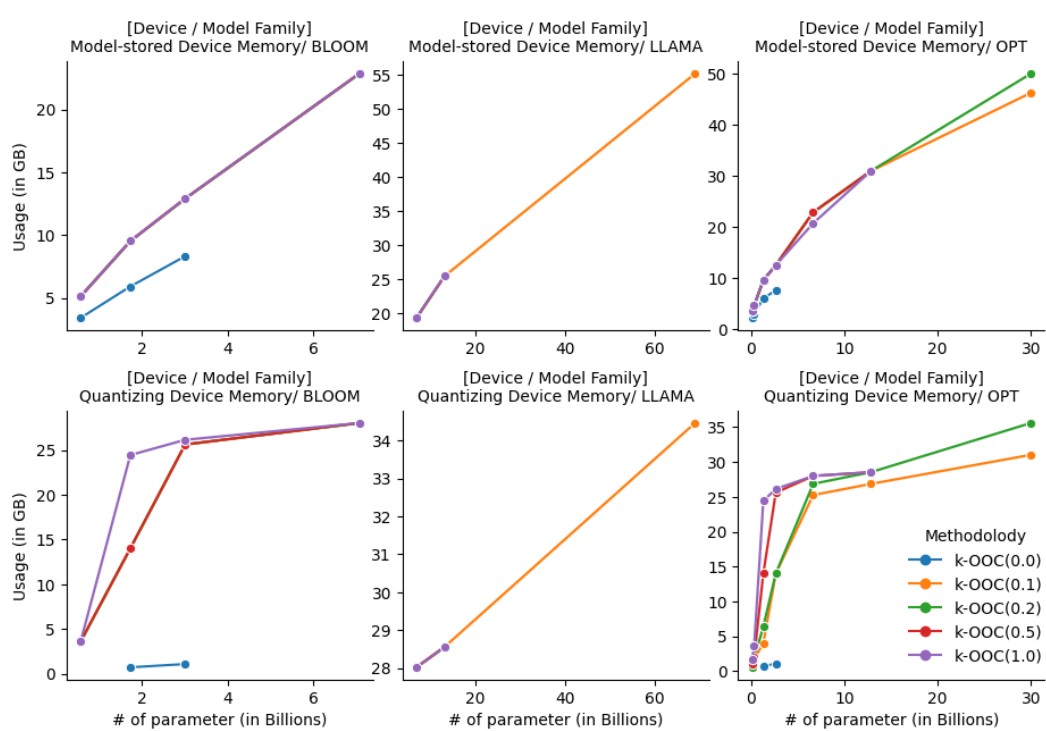

Figure 6: Memory consumption of k-OOC v.s. other GPTQ (k-OOC(0.0)).

| Method | Bit Fam | Eval on | OPT 125M | OPT 350M | BLOOM 560M | OPT 1.3B | BLOOM 1.7B | OPT 2.7B | BLOOM 3B | OPT 6.7B | LLAMA 7B | BLOOM 7B1 | LLAMA 70B |
|---|---|---|---|---|---|---|---|---|---|---|---|---|---|
| GPTQ | 2.1 | WIK V | 216.97 | 234.01 | 78.69 | 45.65 | 35.34 | 32.67 | 23.73 | 16.12 | 39.96 | 15.97 | 6.30 |
| k-OOC(0.1) | 2.1 | WIK V | 177.38 | 234.01 | 68.22 | 42.40 | 34.41 | 31.03 | 23.19 | 15.96 | 39.96 | 15.71 | **6.21** |
| k-OOC(0.2) | 2.1 | WIK V | 201.50 | 234.01 | 65.10 | 44.62 | **31.49** | 29.77 | **22.67** | **15.66** | **17.22** | 15.69 | - |
| k-OOC(0.5) | 2.1 | WIK V | 176.24 | 188.98 | 66.10 | 43.39 | 33.62 | 29.40 | 22.82 | 15.83 | 20.00 | **15.61** | - |
| k-OOC(1.0) | 2.1 | WIK V | **164.97** | **182.50** | **65.05** | **41.59** | 33.25 | **28.67** | 23.06 | 15.90 | - | - | - |
| GPTQ | 2.1 | C4 T | 302.57 | 214.83 | 116.46 | 93.07 | 56.16 | 58.18 | 33.05 | 22.02 | 31.36 | 20.62 | **8.46** |
| k-OOC(0.1) | 2.1 | C4 T | 205.71 | 214.83 | 105.07 | 78.57 | 57.03 | 55.20 | 44.54 | 21.24 | 31.36 | 20.41 | 8.50 |
| k-OOC(0.2) | 2.1 | C4 T | 225.49 | 214.83 | 95.63 | 93.00 | **45.08** | **50.59** | 34.07 | 21.30 | **27.52** | 20.45 | - |
| k-OOC(0.5) | 2.1 | C4 T | 196.10 | 199.42 | 98.25 | 79.45 | 55.45 | 64.91 | 34.05 | **21.01** | 28.56 | **20.40** | - |
| k-OOC(1.0) | 2.1 | C4 T | **180.84** | **176.30** | **93.24** | **73.46** | 49.33 | 55.75 | **31.63** | 21.33 | - | - | - |
| GPTQ | 2.1 | PTB T | 520.55 | **391.55** | 303.53 | 138.64 | 141.01 | 88.13 | 80.18 | 32.00 | 10572.99 | 41.40 | **42.24** |
| k-OOC(0.1) | 2.1 | PTB T | 467.44 | **391.55** | 342.21 | 119.17 | 171.94 | **74.53** | **78.19** | 30.85 | 10572.99 | 40.36 | 61.02 |
| k-OOC(0.2) | 2.1 | PTB T | 603.23 | **391.55** | 301.07 | 121.96 | **112.55** | 78.57 | 79.42 | **29.43** | 13684.60 | 39.91 | - |
| k-OOC(0.5) | 2.1 | PTB T | 465.09 | 431.88 | 318.12 | **116.63** | 135.06 | 79.60 | 82.99 | 29.57 | **8872.44** | 39.32 | - |
| k-OOC(1.0) | 2.1 | PTB T | **428.53** | 498.04 | **232.51** | 119.56 | 115.23 | 81.11 | 78.95 | 30.56 | - | - | - |
| GPTQ | 2.1 | WIK T | 206.00 | 213.67 | 80.48 | 45.54 | 36.00 | 31.70 | 24.21 | 16.42 | 37.52 | 16.73 | 6.94 |
| k-OOC(0.1) | 2.1 | WIK T | 177.65 | 213.67 | 70.31 | 43.48 | 35.85 | 30.83 | 23.74 | 16.28 | 37.52 | 16.52 | **6.87** |
| k-OOC(0.2) | 2.1 | WIK T | 196.89 | 213.67 | **67.36** | 45.19 | **33.13** | 29.49 | 23.63 | **16.07** | 20.39 | 16.51 | - |
| k-OOC(0.5) | 2.1 | WIK T | 161.28 | 182.86 | 68.06 | 43.39 | 35.23 | 28.94 | 23.49 | 16.13 | **18.69** | **16.37** | - |
| k-OOC(1.0) | 2.1 | WIK T | **156.00** | **171.90** | 68.28 | **41.87** | 34.22 | **28.29** | 23.46 | 16.20 | - | - | - |

Table 6: Perplexity of **k-OOC (Ours)** compared against GPTQ and Full Precision (Float 16). It is quantized-trained on the *Wikitext2* dataset and tested on all 3 datasets. The table shows that all variations of k-OOC at least out-performs the GPTQ. The best performance out of selection of "Scan" percentage is in **bold**. Marker "-" indicates the test is not run for that case. To save space, the "Eval on" column uses truncated eval dataset name: "T" stands for "test" and "V" stands for validation. For instance, "WIK T" means "Wikitext2 Test set'.

| Method | Bit Fam | Eval on | OPT 125M | OPT 350M | BLOOM 560M | OPT 1.3B | BLOOM 1.7B | OPT 2.7B | OPT 3B | BLOOM 6.7B | 7B1 | LLAMA 70B |
|---|---|---|---|---|---|---|---|---|---|---|---|---|
| GPTQ | 2.1 | PTB V | 198.16 | 448.65 | 115.95 | 45.30 | 50.67 | 34.92 | 36.33 | 21.70 | 23.79 | **23.57** |
| k-OOC(0.1) | 2.1 | PTB V | 166.47 | 350.78 | 100.74 | **41.88** | 48.94 | 34.92 | **35.32** | 21.41 | **23.48** | 23.57 |
| k-OOC(0.2) | 2.1 | PTB V | 198.16 | **173.92** | **94.71** | 43.89 | **46.96** | 34.92 | 35.57 | 21.12 | 23.54 | - |
| k-OOC(0.5) | 2.1 | PTB V | 170.02 | 437.23 | 104.50 | 43.11 | 48.94 | **34.41** | 35.49 | **20.92** | 23.55 | - |
| k-OOC(1.0) | 2.1 | PTB V | **162.11** | 237.35 | 95.60 | 44.11 | 47.36 | 34.65 | 35.74 | 21.40 | - | - |
| GPTQ | 2.1 | C4 T | 353.58 | 436.71 | 153.45 | 116.00 | 73.06 | 66.79 | 128.52 | 25.07 | 34.28 | **9.41** |
| k-OOC(0.1) | 2.1 | C4 T | 316.33 | 387.90 | **115.71** | 100.04 | 63.22 | 66.79 | 41.08 | 24.34 | 40.68 | 9.41 |
| k-OOC(0.2) | 2.1 | C4 T | 353.58 | **209.86** | 139.66 | **91.50** | **61.70** | 66.79 | 41.55 | **24.05** | 27.57 | - |
| k-OOC(0.5) | 2.1 | C4 T | **295.34** | 347.18 | 138.23 | 94.53 | 62.44 | **56.04** | **39.31** | 24.39 | 24.10 | - |
| k-OOC(1.0) | 2.1 | C4 T | 316.78 | 243.66 | 153.90 | 97.52 | 62.79 | 59.34 | 72.78 | 24.56 | - | - |
| GPTQ | 2.1 | PTB T | 209.82 | 368.59 | 127.87 | 48.16 | 60.02 | 36.31 | 42.79 | 22.01 | 29.05 | **31.79** |
| k-OOC(0.1) | 2.1 | PTB T | 191.50 | 334.35 | 112.64 | **44.42** | 57.76 | 36.31 | 41.21 | 21.65 | 28.66 | 31.79 |
| k-OOC(0.2) | 2.1 | PTB T | 209.82 | **176.33** | **106.68** | 46.52 | **56.20** | 36.31 | 41.31 | 21.47 | 28.82 | - |
| k-OOC(0.5) | 2.1 | PTB T | 186.58 | 352.50 | 112.25 | 45.13 | 56.79 | **35.96** | 41.26 | **21.41** | 28.53 | - |
| k-OOC(1.0) | 2.1 | PTB T | **178.00** | 211.05 | 107.59 | 46.88 | 56.51 | 36.21 | **41.19** | 21.75 | - | - |
| GPTQ | 2.1 | WIK T | 777.08 | 945.00 | 203.12 | 152.08 | 82.30 | 81.62 | 42.42 | 24.53 | 26.71 | **10.28** |
| k-OOC(0.1) | 2.1 | WIK T | 627.71 | 704.28 | 161.72 | **127.33** | 67.29 | 81.62 | 45.75 | 24.30 | 22.65 | 10.28 |
| k-OOC(0.2) | 2.1 | WIK T | 777.08 | **433.39** | **156.98** | 134.20 | **64.08** | 81.62 | 40.64 | **24.17** | 22.70 | - |
| k-OOC(0.5) | 2.1 | WIK T | **488.09** | 778.83 | 158.85 | 142.38 | 71.46 | 71.77 | **38.65** | 25.19 | 22.35 | - |
| k-OOC(1.0) | 2.1 | WIK T | 631.77 | 436.25 | 170.17 | 157.46 | 69.79 | **71.17** | 41.44 | 24.62 | - | - |

Table 7: Perplexity of **k-OOC (Ours)** compared against GPTQ and Full Precision (Float 16). It is quantized-trained on the *PTB* dataset, and tested on all 3 datasets. The table shows that all variations of k-OOC at least out-performs the GPTQ. The best performance out of selection of "Scan" percentage is in **bold**. Marker "-" indicates the test is not run for that case. To save space, the "Eval on" column uses truncated eval dataset name: "T" stands for "test" and "V" stands for "validation". For instance, "WIK T" means "Wikitext2 Test set'.

| OPT Model | | 125m | 1.3B | 2.7B |
|---|---|---|---|---|
| No Matrix-No Group | Default order | - | 374±0 | 91±0 |
| Matrix-Group | Default order | 506±25 | 137±5 | 75±4 |
| | Salient descending | 903±28 | 132±7 | 80±3 |
| | Salient ascending | 2,489±181 | 7,736±441 | 9,974±279 |

Table 8: Perplexity PPL measured for having/ not having Error correction (denoted as "Matrix-Group" and "No Matrix-No Group" correspondingly). See definition in section 1.1. It shows that PPL for having Error correction is lower (better). Measurement for different order of error correction (high salient first v.s. high salient last) is also shown. "Default" (no ranking) v.s. "Salient Descending" is comparable, but "Salient Ascending" degrade the model completely. This explains "Default" order with "Matrix-Group" are selected for the main experimentation in this work.

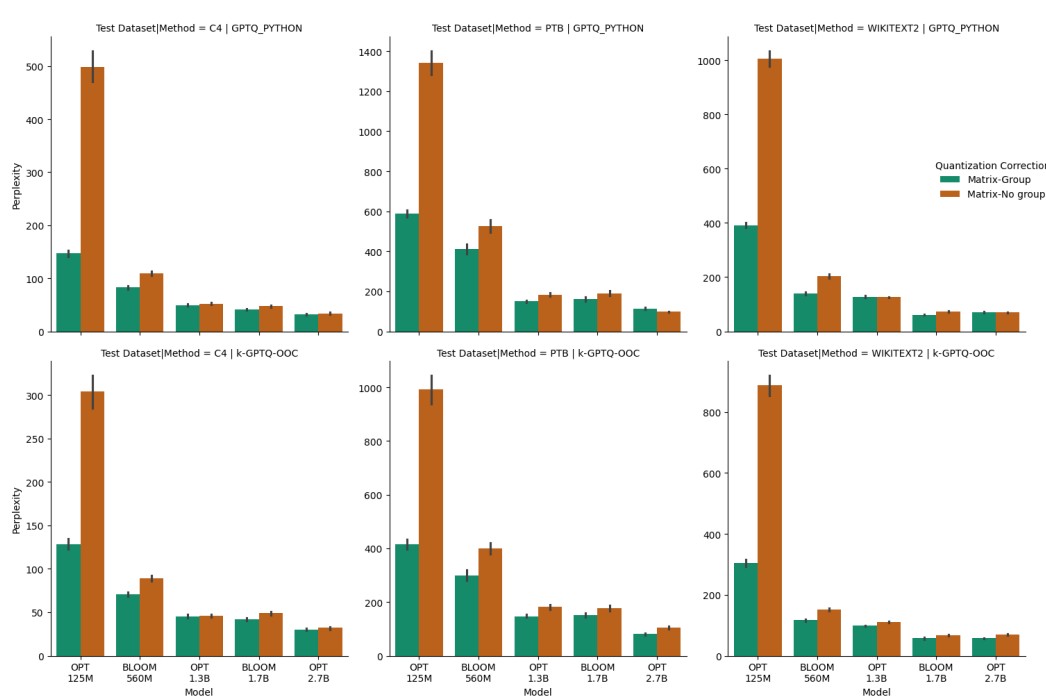

Figure 7: The PPL comparison between "Matrix-Group" and "Matrix-No Group". See definition of each in section 1.1. Matrix-Group outperforms the other option, hence being selected for further experimentation in this work.

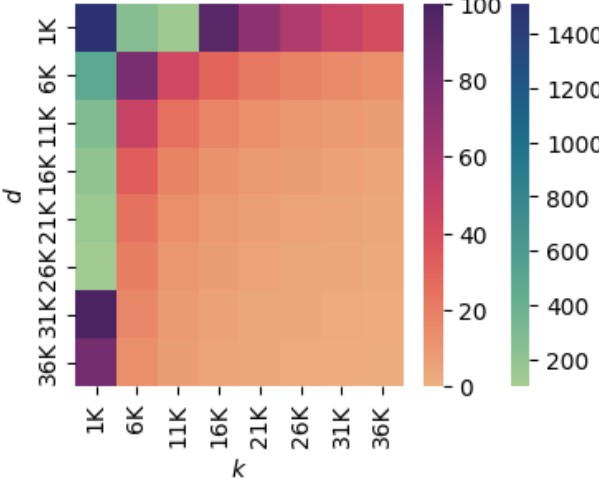

Figure 8: Chunk size $c$ for different $d$ and $k$. The bigger the value of $d$ and $k$, the smaller the chunk size. This is measure on a 80GB VRAM GPU, and the range of $d, k$ covers all model sizes from 125M to 70B.

