# OpenReview forum: "k-Odd One Clear (k-OOC), a novel GPU kernel that improves quantization accuracy and speed of GPTQ algorithm"
_ICLR.cc/2025/Conference — ICLR 2025 Conference Withdrawn Submission_

### Official Review · Reviewer_KLZe · 2024-11-02

**Soundness:** 2
**Presentation:** 2
**Contribution:** 2
**Rating:** 5
**Confidence:** 2

**Summary:**

This paper introduces the k-OOC technique, an optimized GPU kernel that enhances quantization for large language models (LLMs) by extending the existing GPTQ framework. Through a selective high-precision "odd" group quantization and a novel parallel processing scheme, k-OOC aims to improve perplexity (PPL) without significant overhead. The method was tested on multiple LLMs and datasets, yielding improved PPL, especially in smaller models, with notable speed gains in the quantization process.

**Strengths:**

The paper addresses quantization performance both in terms of speed and accuracy, which are crucial for practical deployment of large models.

The novel concept of selectively quantizing "odd" groups with higher precision is interesting and demonstrates a measurable PPL improvement.

The approach leverages GPU-specific parallelism effectively, particularly beneficial for speeding up operations in a way traditional CPU quantization cannot achieve.

**Weaknesses:**

The baseline setup is somewhat limited; comparing only against GPTQ may not fully establish the competitiveness of k-OOC against more sophisticated latest quantization methods

There is a lack of profiling or in-depth kernel performance analysis, such as roofline models, which would help to understand better the efficiency and limitations of k-OOC at the hardware level.

Some statements about GPU occupancy are oversimplified. GPU occupancy depends not just on core usage but also on memory factors like register and shared memory consumption, which aren't explored in depth.

**Questions:**

See weakness

---

### Official Review · Reviewer_R4zW · 2024-11-03

**Soundness:** 2
**Presentation:** 2
**Contribution:** 1
**Rating:** 3
**Confidence:** 3

**Summary:**

This paper focuses on the quantization efficiency of LLM workloads. It provides an optimized GPU kernel design to make the quantization procedure more efficient. It also studies the achieved bit-width of the quantization algorithms besides the bit-width of the weight itself.

**Strengths:**

- It optimizes the quantizing kernel of GPTQ, which makes the quantization procedure more efficient.

**Weaknesses:**

- The achieved accuracy of low-bit quantization in this paper is not good enough. Reducing the quantization overhead is good, it could be better to address the accuracy problem together.
- It lacks the comparison to the state-of-the-art low bit quantization methods, e.g. QuiP. It could be better to show the accuracy under the same bit-width. According to the claim in the recent papers like QuiP, they can achieve better accuracy than this paper.
- It only evaluates the PPL. Better to have some evaluation of the zero shot tasks, e.g., ARC, HellaSwag, PIQA.
- Most of the evaluated models are old. It would be better to evaluate the recent models like Llama 3, Mistral, Qwen2.5, which can better demonstrate the efficacy on the recent workloads.

**Questions:**

- Many existing quantization works do have calculated the effective bit-width of the quantized model, e.g., SqueezeLLM, SpQR. A comparison to them can better illustrate the contribution of this paper.
- It claims the CUDA device can compute 132x32 units simultaneously. However, the number of the blocks to be able to execute simultaneously is also determined by the shared memory and register resource usage.
- There are already many quantization works for extreme low bit-width quantization (e.g., QuiP, and many others). Better to give a comparison to these works. The OPT and Bloom models are old, better use the recent models (Llama 3, Qwen2.5, Mistral, etc). Better to add the evaluation of zero-shot tasks (ARC, PIQA, HellaSwag, etc).

---

### Official Review · Reviewer_nUGr · 2024-11-04

**Soundness:** 2
**Presentation:** 2
**Contribution:** 2
**Rating:** 5
**Confidence:** 3

**Summary:**

The paper presents k-Odd One Clear (k-OOC), a novel GPU kernel designed to enhance quantization accuracy and speed in the GPTQ algorithm for large language models (LLMs). The proposed OOC technique selects an "odd" group to enhance quantization clarity, potentially improving model reasoning capabilities. Additionally, the authors introduce the Bit Family to classify the compression rates of different quantization techniques objectively.

**Strengths:**

- The authors introduce the Bit Family to objectively classify the compression rates of various quantization techniques, providing a consistent metric for calculating the final effective bit precision. This offers a fair basis for comparing different quantization methods.
- The paper proposes a kernel that performs quantization more efficiently and faster than the original GPTQ implementation, showing potential for practical speedups.

**Weaknesses:**

- The paper primarily evaluates model performance using perplexity (PPL). While PPL is a useful metric, its limited improvement in larger models, as the authors note, suggests it may not fully capture the benefits of the proposed method. Including other evaluation metrics, such as MMLU or GSM8K, would provide a more comprehensive assessment of the technique’s effectiveness.
- Although the proposed extension of the GPTQ method improves accuracy, its impact on inference speed is unclear. Unlike weight-only kernels in methods like GPTQ, AWQ, and LUT-GEMM that maintain uniform precision within matrices, the proposed method combines high- and low-precision weights, requiring mixed-precision GEMM operations. To demonstrate real-world applicability, accelerated kernels or system-level optimizations for mixed-precision operations and corresponding latency results are needed.

**Questions:**

Please see the weaknesses section.

---

### Note · Authors · 2025-08-23

I have read and agree with the venue's withdrawal policy on behalf of myself and my co-authors.

---

### Meta-Review · Area_Chair_bsTK · 2024-12-20

**Metareview:**

This work proposes a GPU kernel to improve quantization speed and accuracy for LLMs, offering notable perplexity gains and faster execution. While reviewers agree on its potential, they highlight important shortcomings: limited evaluation metrics (mostly perplexity), old models, and no comparisons with recent methods. This reduces confidence in its broader impact. They also note the need for more thorough hardware-level profiling. The paper show promising ideas, but its scope and comparisons remains incomplete. Without stronger evidence of general applicability, the consensus shifted toward rejection.

**Additional Comments On Reviewer Discussion:**

There was no discussion because the authors did not write a rebuttal.

---

### Decision · Program_Chairs · 2025-01-22

Reject